# Ethical Challenges in Health Care Policy during COVID-19 Pandemic in Italy

**DOI:** 10.3390/medicina56120691

**Published:** 2020-12-11

**Authors:** Davide Ferorelli, Gabriele Mandarelli, Biagio Solarino

**Affiliations:** Interdisciplinary Department of Medicine, Section of Legal Medicine, University of Bari, Piazza Giulio, Cesare 11, 70100 Bari, Italy; gabriele.mandarelli@uniba.it (G.M.); biagio.solarino@uniba.it (B.S.)

**Keywords:** ethical challenges, COVID-19, legal medicine, patient-centered care, clinical management, balancing rights, public health

## Abstract

Since the outbreak of the coronavirus disease 2019 (COVID-19) pandemic, Italy has proven to be one of the countries with the highest coronavirus-linked death rate. To reduce the impact of SARS-CoV-2 coronavirus, the Italian Government decision-makers issued a series of law decrees that imposed measures limiting social contacts, stopped non-essential production activities, and restructured public health care in order to privilege assistance to patients infected with SARS-CoV-2. Health care services were substantially limited including planned hospitalization and elective surgeries. These substantial measures were criticized due to their impact on individual rights including freedom and autonomy, but were justified by the awareness that hospitals would have been unable to cope with the surge of infected people who needed treatment for COVID-19. The imbalance between the need to guarantee ordinary care and to deal with the pandemic, in a context of limited health resources, raises ethical concerns as well as clinical management issues. The emergency scenario caused by the COVID-19 pandemic, especially in the lockdown phase, led the Government and health care decision-makers to prioritize community safety above the individuals’ rights. This new community-centered approach to clinical care has created tension among the practitioners and exposed health workers to malpractice claims. Reducing the morbidity and mortality rates of the COVID-19 pandemic is the priority of every government, but the legitimate question remains whether the policy that supports this measure could be less harmful for the health care system.

## 1. Introduction

The choice to prioritize treatment for those patients who have best chance to survive compared with those who have a higher chance of dying represents one of the issues that are facing even wealthier states since the outbreak of the coronavirus disease 2019 (COVID-19) pandemic. Italy is among the most heavily affected countries worldwide and shows high SARS-Cov-2 coronavirus-linked death rates [1]. Daily bulletins provided by the government informed that thousands of people in Italy were being infected by the novel coronavirus SARS–CoV-2 and many of these patients needed hospitalization with possible admission to intensive care units (ICUs).

Keeping the community morbidity and mortality rates as low as possible is the highest priority of policy makers, but the struggle to keep up with the spreading pandemic and to reduce the pressure on hospitals requires societal as well as health care exceptional measures that also involve ethical issues. The Italian authorities followed the example set by China and imposed a series of measures aimed at guaranteeing social distance, restricted access to public spaces, and stopped non-essential manufacturing sectors as well as business activities during lockdown [2]. The increase in the SARS-COV-2 spread constrains the government to also change the health care policy in order to guarantee the management of COVID-19 symptomatic patients.

From the beginning of the epidemic, the Italian Government has made substantial economic and organizational efforts to guarantee the admission to ICUs, where the percentage of admitted patients ranged consistently between 9% and 11% among those infected [3]. Regional governors have also implemented measures concerning staff resources, hospital beds, and ICU facilities for patients with acute respiratory distress syndrome (SARS) caused by SARS-CoV-2 related pneumonia.

From a regulatory point of view, this was made possible by a series of acts stated by the Council of the Presidency of Ministers, starting from the first Decree issued on 4 March 2020 that contained several measures that focused on the health care system. In this, health personnel shall comply with the appropriate prevention measures for the spread of respiratory infections provided for by the World Health Organization and apply the indications for the sanitation and disinfection of environments provided for by the Ministry of Health. This meant that all elderly persons suffering from chronic or multimorbid diseases or with congenital or acquired immune pressure avoid leaving their home, and persons with symptoms of respiratory infection and fever (over 37.5 °C) were recommended to stay at home and to limit social contacts as much as possible by contacting their doctor. The decree introduced a sort of contact-tracking system for health surveillance managed by general practitioners, a pediatrician, and local public health services for ascertaining people with symptoms suspected of contagion, deciding if they needed to be isolated at home instead of admission to emergency departments.

In this new community-focused care approach, it is further essential to plan and foresee a series of contingencies to manage the increasing number of patients with COVID-19 as well as deal with ethical challenges like initiation or withdrawing/withholding life-sustaining treatments [4,5].

## 2. Issues Relating to “Non-COVID Care”

Outpatient practices and health care services have been substantially limited including ordinary hospitalization and elective surgery that have been cancelled or postponed and this scenario is recurring given the increase in infections.

As a consequence, patients with chronic physical or mental disabilities who need constant care have experienced difficulties at scheduled follow-up in hospitals and in community services [6]. Accordingly, the risk to be infected by direct contact with COVID-19 patients has led most family physicians, who are routinely involved in primary care practices, to reduce or suspend visits except for urgent cases, giving mainly online medical consultations [7,8,9].

In Italy, all citizens have an equal right to access services provided by the National Health System whose health care workers are trained to care for individuals taking into account the patient’s right for information and for shared treatment decision-making [10]. Hence, the routine approach of clinical care is patient-centered, which implies a series of ethical concerns that involve duties to benefit patients, avoid harming them, and maintain professional integrity while acting fairly [11].

A pandemic like COVID-19 represents a catastrophic event that requires a substantial change in the usual health care procedures in order to continue delivering the best level of health care [12]. In a public emergency, the fair allocation of scarce resources available does not allow for the standard of care with respect to saving lives that in normal condition could be saved. This emergency scenario requires a rapid transition to a public-centered approach with the aim to prioritize, in some situations, the community safety above the individual [11,12,13].

The challenging decisions taken in such an emergency phase may also include the need to set criteria for admission to the ICUs in situations of bed shortage [14].

## 3. Coping with an Overwhelmed Health Care System

In a public-centered health care system, the access to life-sustaining treatment could be denied to some citizens when the request for ICUs treatment is beyond the availability of beds. The Italian Society of Anesthesia, Analgesia, Resuscitation, and Intensive Care (SIAARTI) has issued guidelines with respect to the ethical dilemmas lived daily by the practitioners that acknowledge that under exceptional circumstances, the choice to treat only patients with greater chances of therapeutic success may ultimately be justified [15].

The SIIARTI guidelines have been criticized for the ethical implications of such choices, but the difficulty in making medical decisions during a pandemic also remain a great matter of concern for non-life-sustaining treatments [16]. During a public emergency, the risk is to overwhelm the health care system whose disruption could be more harmful for the community than the effects of the COVID-19 pandemic. 

One important issue to consider in this phase is the compliance to the government decisions by the health care workers who are not familiar with patient care in the context of a large-scale, prolonged, public health emergency as are exposed to heightened risk of burnout and compassion fatigue. Therefore it is mandatory for workplace colleagues to support each other and to perform frequent debriefs [4,5,6,7,8,9,10,11,12,13,14,15,16,17]. It is imperative to guarantee the safety of health care workers to minimize their risk of being infected during daily practice, but also ensure that they do not transmit the virus within and outside the hospital [18,19]. These ethical concerns might be more pronounced in practitioners with aged parents or babies and are particularly stressed by the scarce availability of personal protective equipment (PPE), especially in the first phase of the pandemic. Daily counseling by leaders about prevention and protection measures and their rational use can prepare health care to handle such scenarios. The need to guarantee their own physical and mental health care may affect the clinical decision and increase the risk of burnout.

The promotion of health and well-being is a long-standing goal of scientific inquiries [20], however, in such a catastrophic pandemic, we do not have sufficient knowledge about the SARS-Cov-2 disease as well as for related treatment. Most of these medications have been used off label and with compassionate use without previously approved randomized clinical trials (RCT) that support their efficacy and safety [21,22,23], as happens, for example, with the use of investigational drugs such as favipiravir and lopinavir/ritonavir, that can prolong QT interval and cause Torsade de Pointes [24]. Hence, the question arises as to how is it ethically acceptable to treat COVID-19 patients without informed consent, strong scientific evidence, and ethics committee permission with the risk of a malpractice suit. The use of off-label treatments during an emergency involves the physician’s responsibility to carefully balance the risk of a short patient deterioration till death with the ethical interests that have arisen upon these therapies. 

## 4. Informed Consent

Though the special circumstances require an early access to therapies and less restrictive authorization in conducting clinical trials of new therapies, the hazards of using uncontrolled drugs could be associated with serious adverse effects on these patients and could also discourage patients and clinicians from participating in RCTs [25,26]. The risk of harming a single patient during a national health emergency is somewhat inevitable, but it must be mitigated by implementation of measures that are evidence-based and justified by a transparent fair decision-making process [27].

This also includes maintaining high standards in terms of informed consent to treatment and research. With respect to the latter aspect, the risk of therapeutic misconception could be particularly high, given the drive to obtain therapeutic solutions capable of containing the pandemic. Furthermore, we do not believe that it is possible to derogate from the principles usually applied to informed consent, in particular, the duty to inform patients capable of making decisions and to guarantee a free and autonomous choice. The anxiety and pressure associated with the COVID-19 pandemic could push both doctors and patients toward solutions that are not always rational. Some studies have suggested psychologic care guidance, counseling, and social support to health care workers to reduce their physical and mental burden [28]. The most vulnerable patients including those with persistent physical or mental disabilities should not be *a priori* considered as uncapable of making treatment decisions even in the face of choices that would normally not be made in a patient-centered health care context [29,30,31].

## 5. Issues Relating to “Non-COVID Care” That Cannot Be Followed

Many hospitals have been converted into a “Dedicated Care Center” for patients with COVID-19, but they must preserve a standard network for diagnosis and treatment of acute stroke, cerebrovascular and cardiovascular diseases as well as for every medical or surgery emergency [32,33]. The government also must foresee a possible risk of delay in the treatment of non-COVID-19 cases by implementing the efficiency of community services and the availability of non-hospital facilities in terms of disease prevention and screening because most hospital preventive services had been temporary stopped or were limited in activity. This is particularly true for older adults, often confined in nursing homes, with underlying medical conditions such as heart disease, diabetes, or cancer where a delay of primary care can imply morbidity and serious complications up to the risk of death [34,35].

In the context of the COVID-19 pandemic, elderly with many co-morbidities and frail patients need to be correctly informed about the inevitable poor prognosis and shared care planning must be discussed and reviewed with health care professionals [36]. Implementing measures to provide palliative and end-of life care that ensure a comfortable and dignified death is ethically mandatory, although in Italy, the discussion on these issues was only at the beginning, in the period just before the pandemic [37,38].

## 6. Risk of Malpractice Suits

In this emergency scenario, a perception of unfair health care policy and non-ethical allocation of resources can contribute to creating tension among the practitioners and expose health workers and hospital administrators to malpractice claims [39,40]. In Italy, over the last two decades, medical professional liability has become a major concern for health care economics and workers, therefore, since 2007, with the aim of decreasing the risk of harm associated with health care, the Italian government has promoted a program to develop activities related to patient and clinical risk management [41].

Implementing the hospital’s novel care pathways and all proactive measures in response to mitigate COVID-19 spread, starting from the admission of untriaged patients in Emergency Departments up to safety recovery in a dedicated ward will guarantee the standard of care to all citizens, and avoid malpractice being part of the magnitude of indirect health effects due to the pandemic.

## 7. Ethical Resource Allocation Both in COVID and Non-COVID Settings

Balancing rights and public health when a crisis is unfolding is very difficult and the community must be informed by policy-makers with trusted communication. On the other hand, the individual behavior to comply with the government rules is no more important than these challenging decisions [42]. A health care policy based on an ethical framework in every decision-making process is the only way to promote equity of persons and the distribution of risks and benefits in the community. This is even more important in a time of global economic crisis, also due to COVID-19, which makes the role of clinical risk management increasingly central in health care systems. Only balancing between the redefinition of the ethical framework in times of pandemic crisis and the adoption of modern methods of clinical governance, which provide constant cost–benefit analysis and continuous assessment and monitoring of risks, it is possible to promote equity of care and distribution of risks/benefits.

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
