# Peer review of "Ethical Challenges in Health Care Policy during COVID-19 Pandemic in Italy"

_medicina, 2020, doi:10.3390/medicina56120691_

Round 1

Reviewer 1 Report

Review Report

Ethical challenges in healthcare policy during COVID-19 pandemic in Italy

by Ferorelli et al.

The management of the COVID-19 pandemics within the healthcare systems worldwide is associated with new ethical challenges and it is an issue that has to be addressed. The authors pointed at ethical concerns arising from the emergency scenario caused by COVID-19 pandemics in Italy. However, the authors did not give a clear overview which ethical challenges and concerns have been recognized and how these should be (or have been) addressed.

From the manuscript it should be clear which (governmental) measures were carried out and in which extent were the healthcare services limited – e.g. it was written: “ordinary hospitalization and elective surgeries”, but it should be clear how was “ordinary hospitalization” defined.

It was written: “Regional governors also implemented measures concerning staff resources, hospital beds and ICU facilities for patients with acute respiratory distress syndrome (SARS) caused by SARS-CoV-2 related pneumonia”, but it is not clear which “measures concerning staff resources, hospital beds and ICU facilities” were implemented. Also, “From a regulatory point of view, this was made possible with the Decree of the Council of the Presidency of Ministers of 4th March 2020 that contained several measures that focused on the healthcare system” – but this “several measures” were not enlisted.

The authors wrote: “The challenging decisions taken in such an emergency phase may also include the need to set criteria for admission to the ICUs in situations of bed shortage.” I find that it is necessary to recall these criteria and to point out which ethical concerns may arise (or have arisen) as a consequence of their implementation in the clinical praxis.

The Italian Society of Anesthesia, Analgesia, Resuscitation, and Intensive Care (SIAARTI) issued guidelines that faced the ethical dilemma daily lived by the practitioners acknowledging that under exceptional circumstances the choice to treat only patients with greater chances of therapeutic success may ultimately be justified” – It is unclear which ethical dilemma were identified by these guidelines. “The SIIARTI guidelines have been criticized for the ethical implications of such choices…” – it is unclear which “choices” were recommended and what was the criticism.

The majority of issues tackled in the first paragraph on Page 3 (“The first issue…”) hardly fit into the subject of the article. Moreover, the specific points of the addressed issues were missing. E.g. which ethical concerns were raised through adopted “prevention and protection measures”?

The ethical aspects regarding the off-label use of therapeutics and informed consent should be written more clearly. Were there some real-life examples of problems arisen due to the off-label use or troublesome treatment decisions?

In the subheading “5. Issues relating to "normal practice" that cannot be followed” the authors should clearly state which ethical concerns were identified and how were they managed. The information conveyed in the sentence “Implementing measures to provide palliative and end-of life care ensuring a comfortable and dignified death is ethically mandatory although in Italy the discussion on these issues was at the beginning, in a period just before the pandemic [40,41].” is fuzzy.

From the subheading “6. Risk of malpractice suits” it should be clear where is the jeopardy i.e. the biggest risk of malpractice suits. It was written: “Implementing care pathway processes and all proactive measures to mitigate COVID-19 spread and guarantee the standard of care to all citizens…” - It is not clear to which processes the authors point at in order to avoid a malpractice claims.

In conclusion remarks the authors stated that “A Healthcare Policy based on an ethical framework in every decision-making process” – unfortunately, this article does not give an insight on this ethical framework.

Author Response

Reviewer 1

  • From the manuscript it should be clear which (governmental) measures were carried out and in which extent were the healthcare services limited – e.g. it was written: “ordinary hospitalization and elective surgeries”, but it should be clear how was “ordinary hospitalization” defined.

Response 1:

We used the term ordinary hospitalization to define the patients who are admitted in the Hospital as a regular admission instead of non-urgent/emergency access. An ordinary hospitalization could be programmed or performed as a day-hospital. According to the reviewer’s suggestion, we can replace in the text the term ordinary with planned.

In our opinion in paragraphs 2 and 5 we briefly reported the governmental measures that limited the elective and planned healthcare services.

Outpatient practices and healthcare services were substantially limited, including ordinary hospitalization and elective surgery that have been cancelled or postponed

…lead most family physicians, who are routinely involved in primary care practices, to reduce or suspend the visits except for urgent cases, giving mainly on-line medical consultations [7-8-9].

Many hospitals were converted into a “Dedicated Care Center” for patients with COVID-19 but they must preserve a standard network for diagnosis and treatment of acute stroke, cerebrovascular and cardiovascular diseases as well as for every medical or surgery emergency [35-36].

disease prevention and screening because most of the Hospital preventive services are temporary stopped or limited the activity.

In a communication article, we have not enough space for explaining in detail all the measures carried out by the Government mostly because was not the only item that we aimed to analyze.

The measures briefly reported in the text are the mirror of what happened during Lockdown; apart from the admission for acute disease, all the activities were stopped or substantially limited. It was not possible to evaluate exactly the extent of these measures because due to the process of institutional devolution, individual Regions apply different policies.

  • It was written: “Regional governors also implemented measures concerning staff resources, hospital beds and ICU facilities for patients with acute respiratory distress syndrome (SARS) caused by SARS-CoV-2 related pneumonia”, but it is not clear which “measures concerning staff resources, hospital beds and ICU facilities” were implemented. Also, “From a regulatory point of view, this was made possible with the Decree of the Council of the Presidency of Ministers of 4th March 2020 that contained several measures that focused on the healthcare system” – but this “several measures” were not enlisted.

Response 2:

The point underlined by the reviewer is interesting but unfortunately, the answer will not be clear as requested. The “measures concerning staff resources, hospital beds and ICU facilities” were implemented by every region accordingly with local health system resources and organization as well as the number of infections registered that substantially varied from north to south Italy.

The Decree of the Council of the Presidency of Ministers of 4th March 2020 is an eleven pages document that enlisted several measures” that suspended all events in public or private places and give the first guidelines for changing the health care delivery system during Sars Cov-2 Pandemic. 

According to the reviewer comment the paragraph “From a regulatory point of view, this was made possible with the Decree of the Council of the Presidency of Ministers of 4th March 2020 that contained several measures that focused on the healthcare systemwas replaced with the following one: From a regulatory point of view, this was made possible by a series of acts stated by the Council of the Presidency of Ministers, starting from the first Decree issued on 4th March 2020 that contained several measures that focused on the healthcare system. Health personnel shall comply with the appropriate prevention measures for the spread of respiratory infections provided for by the World Health Organisation and apply the indications for the sanitation and disinfection of environments provided for by the Ministry of Health. Was intimate to All elderly persons suffering from chronic or multimorbid diseases or with congenital or acquired immune pressure, to avoid leaving their home; persons with symptoms of respiratory infection and fever (over 37.5° C) were recommended to stay at home and to limit social contacts as much as possible by contacting their doctor. The decree introduces a sort of contact-tracking system for health surveillance managed by general practitioners, a pediatrician, and local public health services for ascertaining people with symptoms suspected of contagion, deciding if they need to be isolated at home instead of admitting to emergency departments.

  • The authors wrote: “The challenging decisions taken in such an emergency phase may also include the need to set criteria for admission to the ICUs in situations of bed shortage.” I find that it is necessary to recall these criteria and to point out which ethical concerns may arise (or have arisen) as a consequence of their implementation in the clinical praxis.
  • The Italian Society of Anesthesia, Analgesia, Resuscitation, and Intensive Care (SIAARTI) issued guidelines that faced the ethical dilemma daily lived by the practitioners acknowledging that under exceptional circumstances the choice to treat only patients with greater chances of therapeutic success may ultimately be justified” – It is unclear which ethical dilemma were identified by these guidelines. “The SIIARTI guidelines have been criticized for the ethical implications of such choices…” – it is unclear which “choices” were recommended and what was the criticism.

Response 3-4:

We thank the reviewer for underlining this point, but the question is that we had no criteria for admission in ICUs need in situations of bed shortage. Italy, as well as other countries, was not prepared to face an emergency like the one by Sars Cov-2 Pandemic. According to the SIAARTI guidelines the Criteria for ICU admission and discharge under exceptional, resource-limited circumstances are flexible for every patient, not only to Covid-19 infected patients. Accordingly, in the text, we reported the criteria suggested by the SIAARTI guidelines under these circumstances to treat only patients with greater chances of therapeutic success. It means that practitioners might ultimately be justified to admit in ICU only people with a chance to survive instead of all citizens who need lifesaving treatment. In our opinion, this issue can raise an ethical dilemma and somewhat criticism by part of the Community.

  • The majority of issues tackled in the first paragraph on Page 3 (“The first issue…”) hardly fit into the subject of the article. Moreover, the specific points of the addressed issues were missing. E.g. which ethical concerns were raised through adopted “prevention and protection measures”?

Response 5:

We substantially agree with the reviewer about this point.

We substitute this paragraph In the Italian healthcare system, during the lockdown, between 8 March 2020 and 18 May 2020, several prevention and protection measures have been adopted. Among these there were already prevention and protection measures of healthcare workers exposed to SARS-CoV-2 [20], but the safety of workers can be further increased by adopting safety procedures that are done in other healthcare systems such as triage for ophthalmic outpatient clinic or systematic screening of healthcare workers with mild symptoms of an acute respiratory tract infection [21-22].

With the following one: These ethical concerns might be more pronounced in practitioners with aged parents or babies and are particularly stressed by the scarce availability of personal protective equipment (PPE) especially in the first phase of the pandemic. Daily counseling by leaders about prevention and protection measures and their rational use can prepare healthcare to handle such scenarios. The need to guarantee their own physical and mental health care may affect the clinical decision and increase the risk of burnout.

  • The ethical aspects regarding the off-label use of therapeutics and informed consent should be written more clearly. Were there some real-life examples of problems arisen due to the off-label use or troublesome treatment decisions?

Response 6:

We agree with the reviewer that the issue about off-label prescriptions during pandemic deserve more attention but this is a communication article so that is not possible to deep every item as well as we like. However, the paragraph was substantially revised.

OLD

The promotion of health and well-being is a long-standing goal of scientific inquiries [23] hence the question arise about how is it ethically acceptable to treat COVID-19 patients with off label and compassionate use therapies without previously approved randomized clinical trials (RCT) that support their efficacy and safety [24-25-26], as happens, for example, with the use of investigational drugs such as favipiravir and lopinavir/ritonavir, that can prolong QT interval and cause Torsade de Pointes [27].

NEW

The promotion of health and well-being is a long-standing goal of scientific inquiries [23] however in such a catastrophic pandemic we had not sufficient knowledge about SARS-Cov-2 disease as well as for related treatment. Most of these medications were used off label and with compassionate use without previously approved randomized clinical trials (RCT) that support their efficacy and safety [24-25-26], as happens, for example, with the use of investigational drugs such as favipiravir and lopinavir/ritonavir, that can prolong QT interval and cause Torsade de Pointes [27]. Hence the question arises about how is it ethically acceptable to treat COVID-19 patients without informed consent, strong scientific evidence, and ethics committee permission with the risk of a malpractice suit. The use of off-label treatments during an emergency involves the physician's responsibility to carefully balance the risk of a short patient’s deterioration till death with the ethical interests arisen upon these therapies.

  • In the subheading “ Issues relating to "normal practice"that cannot be followed” the authors should clearly state which ethical concerns were identified and how were they managed. The information conveyed in the sentence “Implementing measures to provide palliative and end-of life care ensuring a comfortable and dignified death is ethically mandatory although in Italy the discussion on these issues was at the beginning, in a period just before the pandemic [40,41].” is fuzzy.

Response 7:

In our opinion is understandable that the delay in treatment of no covid-19 patients could identify an ethical concern about the risk to neglect their health. The issue of end-of-life decisions and directive advances in Italy is still a matter of concern.

  • From the subheading “ Risk of malpractice suits” it should be clear where is the jeopardy i.e. the biggest risk of malpractice suits. It was written: “Implementing care pathway processes and all proactive measures to mitigate COVID-19 spread and guarantee the standard of care to all citizens…”- It is not clear to which processes the authors point at in order to avoid a malpractice claims.

Response 8:

According to the reviewer's observation, we will change the following paragraph: “Implementing care pathway processes and all proactive measures to mitigate COVID-19 spread and guarantee the standard of care to all citizens will avoid that malpractice will be part of the magnitude of indirect health effects due to the COVID-19 outbreakwith the present one: Implementing Hospital’s novel care pathways and all proactive measures in response to mitigate COVID-19 spread, starting from the admission of untriaged patients in ED up to safety recovery in a dedicated ward will guarantee the standard of care to all citizens, and avoid that malpractice will be part of the magnitude of indirect health effects due to the Pandemic.

Reviewer 2 Report

General comments

This article discusses the COVID-19 pandemic in Italy and reviews the ethical challenges raised by Government decisions intended to rein in the pandemic. The topic is important and timely. The article is a useful review and summary of the pandemic in Italy and the ethical issues that arise. The main shortcoming of the article is that the ethical issues and challenges are merely pointed out and neither analysed nor discussed in any detail. The authors make a few reasonable, normative statements on these issues, but there is little or no attempt to justify them. It is nonetheless useful to see what ethical challenges arise and in what context. 

Specific comments

Line 29: The title of the first section is “Introduction to conceptual analysis”. I would suggest shortening the title to “Introduction”, because I don’t think there is any conceptual analysis in this article.

Line 43: “constraint” should be “constrains”.

Line 68: “imply” should be “implies”.

Line 90: “The first issue to consider in this phase ...” there is no second issue. I suggest rephrasing this, for example with “One issue to consider ...” or “One important issue to consider ...”.

Line 94: “Firstly, ...” there is no “secondly”.

Line 128: I don’t understand this section title, what does it mean that “normal practice” cannot be followed? Limited availability of regular clinical care/health services?

Lines 131–135: I don’t understand this sentence. Should “implementing” be “maintaining”? The grammar and structure of the sentence requires some work.

Lines 157–158: “... is no more important than ...” really? Is this supposed to be “... is no less important than ...” or “... is just as important as ...”? If not, I think the authors need to expand and clarify this point.

Author Response

Reviewer 2

  • Line 29: The title of the first section is “Introduction to conceptual analysis”. I would suggest shortening the title to “Introduction”, because I don’t think there is any conceptual analysis in this article.

Thanks, as you suggested we have changed the title of the paragraph.

  • Line 43: “constraint” should be “constrains”.

We corrected as you suggested.

  • Line 68: “imply” should be “implies”.

We corrected as you suggested.

  • Line 90: “The first issue to consider in this phase ...” there is no second issue. I suggest rephrasing this, for example with “One issue to consider ...” or “One important issue to consider ...”.

We corrected as you suggested.

  • Line 94: “Firstly, ...” there is no “secondly”.

We deleted Firstly as you suggested.

  • Line 128: I don’t understand this section title, what does it mean that “normal practice” cannot be followed? Limited availability of regular clinical care/health services?

Thanks for this tip. It was actually not clear. We meant care not related to Covid and therefore of non-Covid patients. For this reason we have replaced "normal practice" with "non-COVID care".

  • Lines 131–135: I don’t understand this sentence. Should “implementing” be “maintaining”? The grammar and structure of the sentence requires some work.

We mean that since Covid cases determine the risk of delay in the treatment of non-COVID-19 cases, the Government must implement the efficiency of Community services and the availability of non-Hospital facilities also in terms of disease prevention and screening because most of the Hospital preventive services are temporary stopped or limited the activity

  • Lines 157–158: “... is no more important than ...” really? Is this supposed to be “... is no less important than ...” or “... is just as important as ...”? If not, I think the authors need to expand and clarify this point.

Thank you very much for this review. We actually made a mistake. We corrected as you suggested with " is just as important as”.

Reviewer 3 Report

I think this is a very thoughtful exploration of the ethical challenges in health care policy during the pandemic in Italy, with major implications for other countries where similar policies have been pursued. My only comment is that the English needs detailed attention, preferably by a native speaker , to ensure that meaning is crystal clear.  eg final sentence of abstract should be rephrased "...but the legitimate question remains whether the Policy that supports this measure could be less harmful for the health care system."

Author Response

I think this is a very thoughtful exploration of the ethical challenges in health care policy during the pandemic in Italy, with major implications for other countries where similar policies have been pursued. My only comment is that the English needs detailed attention, preferably by a native speaker , to ensure that meaning is crystal clear.  eg final sentence of abstract should be rephrased "...but the legitimate question remains whether the Policy that supports this measure could be less harmful for the health care system."

Response: Thanks a lot for your review. We did what you suggested us and replaced the final sentence of the abstract with "but the legitimate question remains whether the Policy that supports this measure could be less harmful for the health care system".

Round 2

Reviewer 1 Report

Review Report

Ethical challenges in healthcare policy during COVID-19 pandemic in Italy

by Ferorelli et al.

The management of the COVID-19 pandemics within the healthcare systems worldwide is associated with new ethical challenges and it is an issue that has to be addressed. The authors pointed at ethical concerns arising from the emergency scenario caused by COVID-19 pandemics in Italy.  

Unfortunately, due to the writing style, the informational content of the manuscript remains imperceptible. A good rule of thumb is to keep the average sentence length around 20-25 words. There are many too long sentences, where, upon reaching the end, the reader cannot remember how the sentence began.

I find that the author should keep the following in mind: “Vigorous writing is concise. A sentence should contain no unnecessary words, a paragraph no unnecessary sentences, for the same reason that a drawing should have no unnecessary lines and a machine no unnecessary parts. This requires not that the writer make all his sentences short, or that he avoids all detail and treat his subjects only in outline, but that he makes every word tell.” (Strunk and White, The Elements of Style).

I suggest authors to read the following article:

https://www.enago.com/academy/how-to-optimize-sentence-length-in-academic-writing/

The authors should, at least, rewrite the following marathon sentences: Lines 38, 56, 60, 116, 126, 147, 164, 169. These are source of many confusions. Moreover, these sentences are fuzzy: line 122, 56 (“was intimate to”). Moreover, sentences where at least one comma is not missing are rare.

In not just few instances (e.g., line 55: apply the indication (vs. recommendations) […] environments (vs. rooms), to name just a two), are indeed used English words, but the words are wrong.

I find that readers would benefit, if the authors would enlist the recognized ethical concerns at the beginning of the article.